# Trigeminal Schwannoma Presenting with Masticatory Muscle Atrophy: A Rare Case Report

**Tona Yoshino, Keiji Shinozuka \*, Kana Yokoyama, Ayana Yamakawa and Morio Tonogi**

Department of Oral and Maxillofacial Surgery, Nihon University School of Dentistry, Tokyo 101-8310, Japan
\* Correspondence: shinozuka.keiji@nihon-u.ac.jp; Tel.: +81-3-3219-8082

**Abstract:** Trigeminal schwannoma (TS) presenting with masticatory muscle atrophy is very rare. Here, we report the case of an 80 year-old male patient with TS presenting with masticatory muscle atrophy in October 2014. The patient had been diagnosed with hypertrophic pachymeningitis and right intracranial TS in 2013 by his neurologist, for which he was treated with steroids. However, his symptoms in the trigeminal innervation region persisted. He visited our department due to difficulty in opening his mouth as well as eating. Surgery was not performed owing to his advanced age and general condition. His trismus was improved by mouth opening training via manual manipulation for three years. We suggest that TS should be considered in the differential diagnosis of patients with masticatory muscle atrophy. Consultation with a neurologist is also essential. Manual manipulation may be an effective non-surgical treatment option for patients with difficulty in mouth opening owing to TS.

**Keywords:** hypertrophic pachymeningitis; manipulation; masticatory muscle atrophy; trigeminal schwannoma; case report

## 1. Introduction

Trigeminal schwannoma (TS) is exceedingly rare, accounting for only 0.07–0.36% of all intracranial tumors and only 0.8–8% of all intracranial schwannomas [1]. TS tends to occur in middle-aged patients and is more common among women than among men [1]. Intra-cranial tumors cause various neurological symptoms in the oral and maxillofacial area. Patients with TS may be asymptomatic for an average of up to three years until the mass grows and compresses adjacent structures, leading to pain, dysarthria, dysphagia, or trismus [2]. However, most TS patients visit neurologists because they experience neurological symptoms in early-stage TS. To our knowledge, a case of TS has not been previously reported by a dentist.

Here, we report a case of masticatory muscle atrophy in the right trigeminal innervation region, which was attributed to TS. Consequently, the patient's dentist referred him to a neurologist.

## 2. Case Report

In October 2014, an 80 year-old male patient with difficulty in opening his mouth was referred by his primary dentist to the Department of Oral Surgery at [X]. His medical history included known diagnoses of sinusitis, bilateral hearing loss, reflux esophagitis, lipid metabolism disorder, benign prostatic hyperplasia, hypertension, hypertrophic pachymeningitis, and right intracranial TS.

Clinical extra-oral examination revealed facial asymmetry and atrophy of the right masseter muscle (Figure 1), which was first observed in December 2012 by the primary care dentist before the patient presented to our department. However, the patient did not report any disorientation, trauma, dysphagia, or dysphonia. Intra-oral examination revealed normal occlusion with restricted mouth opening (20 mm) and associated mandibular deviation to the right. There were no signs of inflammation or infection.

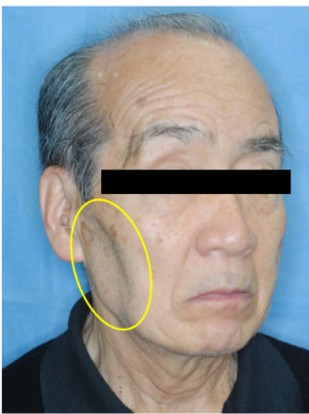

**Figure 1.** Atrophy of the right side of the masseter muscle.

Panoramic radiographic findings were inconclusive for the presenting clinical symptoms (Figure 2). Computed tomography (CT) showed atrophy of the right masseter muscle, temporalis muscle, and lateral and medial pterygoid muscles (Figure 3). Blood tests were normal, and the patient tested negative for antinuclear antibody and rheumatoid factors. Physiological test results revealed bilateral hearing loss and paresthesia of the right side of the face. Surface electromyography recorded no action potentials in the right masseter and temporalis muscles (Figure 4).

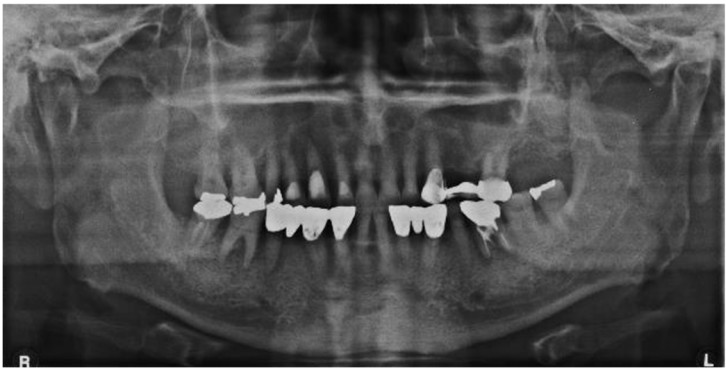

**Figure 2.** Panoramic radiography: no abnormal findings related to the chief complaint were noted.

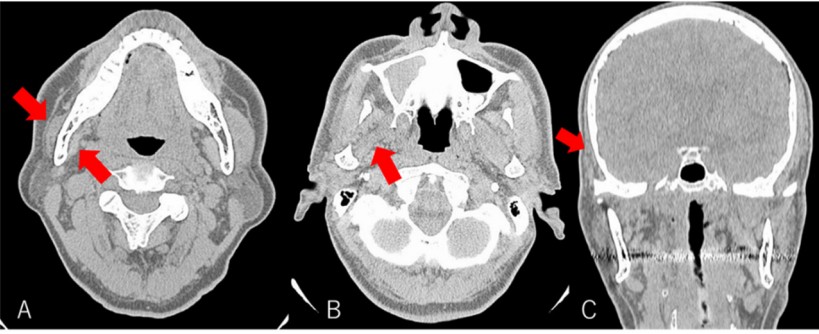

**Figure 3.** Axial contrast-enhanced computed tomography (CT) of the soft tissue (**A**,**B**) and coronal CT of the head (**C**). Atrophy of the right masseter muscle, lateral and medial pterygoid muscles, and temporalis muscle (red arrow) compared with the contralateral side.

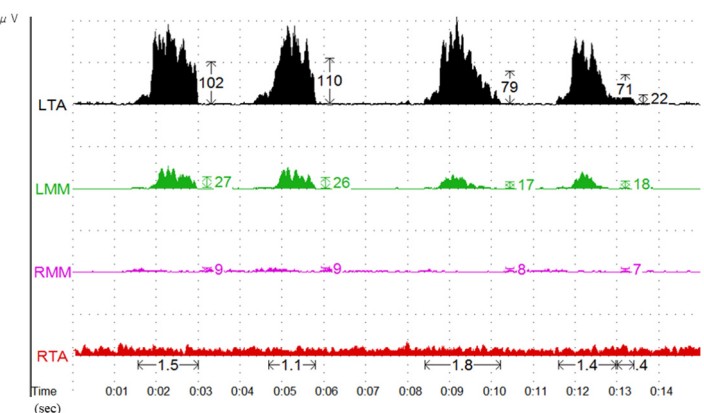

**Figure 4.** Surface electromyography: Action potential was not recorded from the right masseter and temporal muscles. Left anterior temporalis (LTA), left masseter (LMM), right masseter (RMM), right anterior temporalis (RTA). *X*-axis: time, *Y*-axis: EMG activity (µV).

Our consultation with the neurologist revealed that the patient had a known medical history of hypertrophic pachymeningitis and TS. In March 2013, paresthesia and dysgeusia in the right trigeminal nerve innervation area were detected. In addition, the patient experienced difficulty in eating. In April 2013, he visited the Department of Neurosciences at [XX] and was hospitalized for dietary management and intensive examination and treatment. Head magnetic resonance imaging (MRI) was performed, and hypertrophic pachymeningitis and TS were suspected. No osteoarthritic change of the condyle and no articular disc displacement on both sides of the TMJ was found on multiplanar reconstructed MR images (Figure 5).

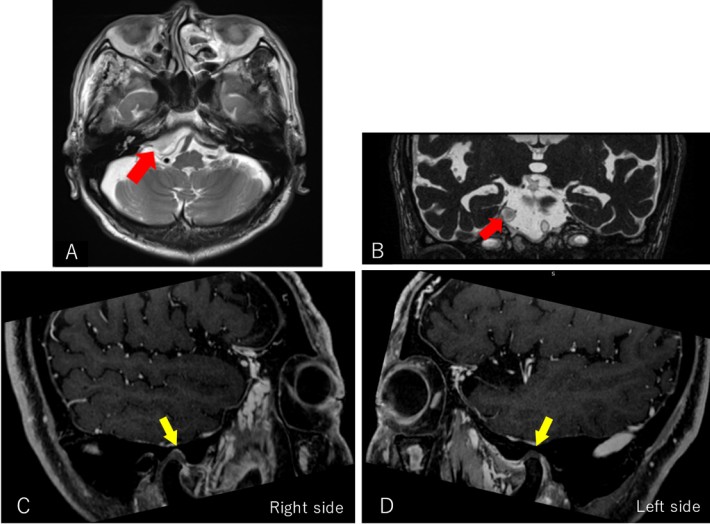

**Figure 5.** Axial (**A**) and coronal (**B**) T2-weighted magnetic resonance imaging showing a heterogeneously enhanced mass (red arrow). Reconstructed MR images showing no abnormal structure of the TMJ components. (**C,D**) The articular disc (yellow arrow) is in a correct position.

Hypertrophic pachymeningitis associated with vasculitis was diagnosed based on inflammatory findings. Blood examination results revealed a high erythrocyte sedimentation rate. Myeloperoxidase anti-neutrophil cytoplasmic antibody (MPO-ANCA) was also positive. The diagnosis of hypertrophic pachymeningitis was further supported by dura mater biopsy findings. Granulomatosis with polyangiitis (GPA) was also suspected due to the patient's sinusitis and hearing loss. However, no findings consistent with GPA were noted on nasal mucosal biopsy.

The patient required urgent treatment due to symptom progression. Steroid pulse therapy with methylprednisolone was administered at 1000 mg/day for 3 days. This was initiated at 60 mg and gradually tapered. MPO-ANCA was negative, and no inflammatory response was observed following steroid pulse therapy. However, the patient's symptoms in the trigeminal innervation region persisted.

In October 2014, the patient visited our hospital because of difficulty in opening his mouth and troubled eating. We consulted his neurologist, who observed persisting lesions on MRI. Therefore, we determined that the muscle atrophy was owing to intracranial TS rather than hypertrophic pachymeningitis. Owing to the patient's age and medical history, and upon consultation with his neurologist, the patient decided against surgery. We subsequently performed radiography and electromyography. The patient performed exercise therapy by placing his thumb against the upper anterior teeth and index finger against the lower anterior teeth and then forcefully separating his upper and lower jaws until maximum mouth opening distance was achieved. He held the stretched muscle for 30 s, and then relaxed for 30 s (1 cycle); this was repeated for three cycles per session for five sessions per day.

Although the right-sided mandibular deviation during mouth opening persisted, the degree of opening improved to >40 mm, and the patient's eating disorder disappeared entirely by 2015 (Figure 6). Annual CT conducted from 2015 to 2020 revealed no abnormal findings or progression in muscle atrophy.

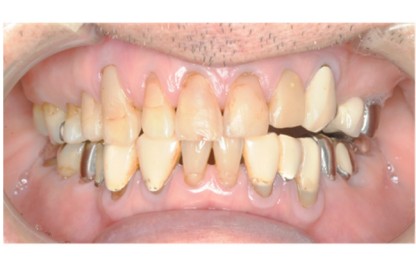 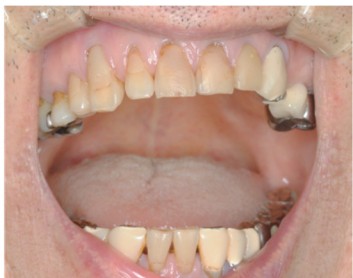

**Figure 6.** The degree of mouth opening improved to >40 mm, while the right-side mandibular deviation observed during mouth opening persisted.

## 3. Discussion

TS generally arises from Schwann cells of the sensory root and can originate from any section of the fifth cranial nerve. Correspondingly, various symptoms and signs may develop. In general, patients with TS tend to present with symptoms, including paresthesia of the face, facial pain, and tinnitus. Nevertheless, the tumor may also be asymptomatic, as it often develops in the Meckel cave, where blood vessels and nerves are sparse. Compared to paresthesia of the face, movement disorders, including muscle atrophy and deviation of the jaw, often occur as the condition progresses and indicate advanced-stage tumor [1].

Differential diagnoses of masticatory muscle atrophy include pure trigeminal motor neuropathy [3], progressive facial hemiatrophy, and scleroderma [4]. Pure trigeminal motor neuropathy is characterized by muscle atrophy in the dominant region of trigeminal nerve innervation, with sensory disturbances being absent. Although the cause of this disease is unknown, viral infection has been proposed as a possible cause. Progressive facial hemiatrophy, also known as Parry–Romberg syndrome, is a cranio-facial disorder characterized by progressive shrinking and deformation of one side of the face, with atrophy of the subcutaneous connective and fatty tissues. The underlying pathogenesis is not well understood. Congenital mechanisms, disturbance of fat metabolism, and trophic malformation of the cervical sympathetic trunk have been implicated [5]. Scleroderma is an autoimmune disease, and most affected patients tend to test positive for the rheumatoid factor. It is also not associated with sensory-related symptoms [4].

In our case, (1) there was no oral or facial lesion, trigeminal neuralgia, or facial paralysis; (2) the time of onset of difficulty in mouth opening coincided with the onset of

TS; (3) paresthesia was noted; and (4) test results for antinuclear antibody and rheumatoid factor were negative. Based on these findings, we concluded that masticatory muscle atrophy most likely occurred because of TS.

TS is usually benign and is generally curable through complete surgical removal. However, for asymptomatic or small lesions, the best approach may be patient observation [6]. Upon consultation with his neurologist, our patient decided on the latter approach and declined surgery due to his age, general condition, and quality of life.

Manual manipulation has been used to treat patients with limitations in mouth opening caused by temporomandibular disorders. Such disorders may be attributable to masticatory muscle, temporomandibular joint, and central nervous system disorders [7]. However, the curative effect of manual manipulation varies between studies; thus, the necessity for manual manipulation remains controversial. Nagata et al. [7] performed a randomized controlled trial examining the efficacy of manipulation and found an advantage of manipulation only during the first treatment session. Manual therapy to improve difficulties in mouth opening may be effective during the early stages of TS. In this case, it is likely that treatment with manual manipulation in the early stages improved the patient's ability to open his mouth.

In conclusion, masticatory muscle atrophy, particularly atrophy caused by TS, is rare. To the best of our knowledge, this is the first study to report TS as described by a dentist, which should be considered as part of the differential diagnosis of masticatory muscle atrophy. This approach should acknowledge the importance of consultation with specialists. In previously mentioned cases, the complete removal of tumors did not improve the patient's condition [6]. Moreover, as the population ages, the number of patients who cannot undergo surgery may increase. Although non-surgical treatment of masticatory muscle atrophy has not been firmly established, our results suggest that manual manipulation of the mouth is a promising treatment for patients with difficulties in opening their mouth owing to TS.

**Author Contributions:** Conceptualization, K.S.; methodology, K.S. and M.T.; investigation, T.Y., K.Y. and A.Y., data curation, T.Y., K.Y. and A.Y., writing and original draft preparation, T.Y.; writing, review, and editing, K.S. and M.T.; funding acquisition, K.S.; visualization, T.Y. and K.S.; supervision, K.S. and M.T.; project administration, M.T. All authors have read and agreed to the published version of the manuscript.

**Funding:** This research was funded by a Grant-in-Aid for Scientific Research C (no. 20K10126) from the Japan Society for the Promotion of Science and grants from the Sato Fund (2018, 2020), Nihon University School of Dentistry.

**Institutional Review Board Statement:** Not applicable.

**Informed Consent Statement:** Written informed consent has been obtained from the patient to publish this paper.

**Data Availability Statement:** Data sharing is not applicable.

**Conflicts of Interest:** The authors declare no conflict of interest.

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
