# Peer review of "Trigeminal Schwannoma Presenting with Masticatory Muscle Atrophy: A Rare Case Report"

_applsci, doi:10.3390/app12199919_

Round 1
Reviewer 1 Report
I enjoyed reading the report on an interesting case. This case is quite unique, and is considered to be a rare case in the oral and maxillofacial area.
I have some questions to the authors.
Although TS is presumed to be the main cause of masticatory muscle atrophy, and the mandibular deviation to right side is also thought to be due to it, evaluation of the right articular disc may be necessary. In addition, his right mandibular condyle also showed erosive finding in panoramic radiograph (Figure 2). Therefore, it is necessary to comment the status of the his condyles and his articular disc in TMJ.
Please add some comments in the manuscript about the status of the patient's condyles and articular disc.
Reviewer 2 Report
Congratulation to this fine written case-report.
